# INTERPRETING DEEP CLASSIFICATION MODELS WITH BAYESIAN INFERENCE

## ABSTRACT

In this paper, we propose a novel approach to interpret a well-trained classification model through systematically investigating effects of its hidden units on prediction making. We search for the core hidden units responsible for predicting inputs as the class of interest under the generative Bayesian inference framework. We model such a process of unit selection as an Indian Buffet Process, and derive a simplified objective function via the MAP asymptotic technique. The induced binary optimization problem is efficiently solved with a continuous relaxation method by attaching a Switch Gate layer to the hidden layers of interest. The resulted interpreter model is thus end-to-end optimized via standard gradient back-propagation. Experiments are conducted with two popular deep convolutional classifiers, respectively well-trained on the MNIST dataset and the CIFAR10 dataset. The results demonstrate that the proposed interpreter successfully finds the core hidden units most responsible for prediction making. The modified model, only with the selected units activated, can hold correct predictions at a high rate. Besides, this interpreter model is also able to extract the most informative pixels in the images by connecting a Switch Gate layer to the input layer.

## 1 INTRODUCTION

Deep Neural Networks (DNNs) have achieved state-of-the-art performance in various machine learning tasks, such as classification and regression. However, they are mostly like a black box and it is not clear how a specific decision is made by the model. Pursing interpretability of deep classification models would promote their research and application in many aspects such as error analysis and model refinement. Moreover, interpreting the prediction is also beneficial to deep learning model users, e.g. in rationalizing medical diagnosis. Recently, several studies (Shwartz-Ziv and Tishby, 2017; Koh and Liang, 2017; Li and Yuan, 2017; Paul and Venkatasubramanian, 2014) attempt to make DNN models more transparent from the perspective of model interpretability, by proposing measures to investigate the learning procedure, or by building analytical mathematic models for the model structure. Besides, there are also some recent works interpreting deep models from the perspective of prediction explanation (Ribeiro et al., 2016; Lundberg and Lee, 2017; Alvarez-Melis and Jaakkola, 2017; Bach et al., 2015; Lei et al., 2016), which shed light on particular predictions via constructing models for local approximation or end-to-end rationalizing.

One of the most frequently asked questions for interpreting DNNs is what hidden units learn from global optimization. Some researchers attempt to give an answer via visualizing units in hidden layers of a deep neural network (Simonyan et al., 2013; Zeiler and Fergus, 2014; Selvaraju et al., 2016). For example, Simonyan et al. (2013) proposed two techniques for visualization based on the gradient of the class scores w.r.t. inputs. Without computing gradients, Zeiler and Fergus (2014) introduced a method to visualize feature maps in hidden layers with a Deconvnet, which maps the activation in a hidden layer back to the input pixel space.

Despite the efforts devoted to interpreting deep models, the previous research still focuses on analyzing the sensitiveness of a model rather than explicitly explaining how each hidden unit contributes to the final decision. In this paper, we propose a novel interpreter to approach the interpretability of a well-trained classification model. In particular, we investigate effects of the units on prediction making for each data class by finding the core units in hidden layers that are most responsible for making the prediction. The model with its core units activated while all others masked can still

make decisions with a high accuracy. Such a process of units selection can be modeled as an Indian Buffet Process (IBP) (Ghahramani and Griffiths, 2006), and we propose to solve it with the Bayesian inference framework. With the MAP asymptotic technique (Broderick et al., 2013), we derive a simplified objective function that is easy to optimize, which combines two terms: the number of selected units and the prediction accuracy. We then solve the binary optimization problem with the continuous relaxation by adding a regularization term in the derived objective function. The continuous relaxation is realized by connecting a Switch Gate layer, in which each unit serves as an ON/OFF controlling gate, to the hidden layer of interest. After adding the switch gate, the model becomes continuous and can be optimized through back-propagation. Then we can obtain the core units to interpret the trained model.

Via global end-to-end optimization, the proposed interpreter provides the least number of necessary units that preserve most information for making the classification decision. Through extensive experiments, we demonstrate that our method can obtain the most responsible units, i.e., the core units, not only in a single layer but also in a series of layers. Our main contributions are three-fold: 1) We propose a novel and general method to interpret a well-trained classification model via discovering core units in hidden layers for prediction making. 2) We formulate the unit selection as a generative IBP process, and derive a simplified objective function through asymptotic Bayesian inference. 3) We apply the effective continuous relaxation method, and introduce a Switch Gate layer and a center-based binary classifier, to solve the apparently complex binary optimization problem.

## 2 RELATED WORK

To date, there have been considerable research works that address the interpretability of deep neural networks. A broad overview about the interpretability of machine learning is provided by Doshi-Velez and Kim (2017). Several recent studies attempt to interpret deep neural networks through accounting for the architecture of a model (Shwartz-Ziv and Tishby, 2017; Koh and Liang, 2017; Li and Yuan, 2017; Paul and Venkatasubramanian, 2014). In Shwartz-Ziv and Tishby (2017), a deep neural network is regarded as the information bottleneck. Based on this work, during optimization, hidden layers firstly focus on capturing information from both input data and corresponding labels. At the last period, hidden layers try to capture more information from labels and also to get rid of irrelevant information extracted from input data as much as possible in order to shrink models. In the work of Koh and Liang (2017), influence functions are used to measure the extent to which optimization state and predictions are changed in terms of a small perturbation on the training set.

There are also some works interpreting deep models via explaining particular predictions (Ribeiro et al., 2016; Lundberg and Lee, 2017; Alvarez-Melis and Jaakkola, 2017; Bach et al., 2015). The work of Ribeiro et al. (2016) approximates the prediction of a network locally in a faithful way with a simple and interpretable model. Their methodology is similar to the classical local-linearization method, which discovers an unknown system through locally approximating. Alvarez-Melis and Jaakkola (2017) attempted to explain the predictions of a sequence-to-sequence model via a causal framework, which infers causal dependencies between original input and output tokens using perturbed input-output pairs.

The methodology of this paper is a combination of model interpretability and prediction interpretability. We propose a novel and general method to investigate the effects of the units in hidden layers for each data class. The selection of the most responsible units are not based on particular predictions, but on one whole class of predictions. For different classes, we use the same structure of the model for binary optimization.

## 3 METHOD

In this section we provide details of the proposed interpreter. We investigate the contributions of the units in a hidden layer to predicting an input as a specific class. Units operate like information channels, through which useful information from the input end is transferred to the output end. For predicting an input as the class of interest, we are curious about which units are most responsible for the prediction making. In other words, the model with other units masked can still make correct predictions. We aim to select such a set of units, called the core units, in a hidden layer for a specific training dataset. To this end, we model such a process of units selection as an Indian

Buffet Process (Ghahramani and Griffiths, 2006), considering the superiority of a generative model in interpretability.

## 3.1 PROBLEM FORMULATION AND OBJECTIVE FUNCTION

We assume that a DNN classifier $F(\cdot)$ is well-trained on a dataset $D = \{(x_n, y_n)\}_{n=1}^N$, where $x_n \in \mathbb{R}^d$ represents the $d$-dimensional input and $y \in \{1, 2, \ldots, H\}$ represents labels from $H$ categorical classes. The output of the classifier $y_n^f = F(x_n)$ is an $H$-dimensional probability vector, which contains prediction scores for each class. The input $x_n$ is classified into the class $p_n$, which gives the maximum prediction scores. For investigating the effects of the units on prediction for the class of interest, $c$, we assign a new label, denoted as $y_n'$, to each input $x_n$ as follows. If the classifier $F(\cdot)$ predicts the input to be class $c$ (i.e., $p_n = c$), label 1 will be assigned to $y_n'$; otherwise label 0 will be assigned. For discovering the core units responsible for making prediction as class $c$, we construct a new dataset consisting of "positive" and "negative" data, i.e., inputs predicted as class $c$ and those not. We denote this dataset as $D_o^c$. It contains all inputs with new label 1 and the same number of vectors with new label 0 randomly selected, i.e., $D_o^c = \{(x_n, y_n')\}_{n=1}^{N'}$, where $N'$ is the total number of selected inputs.

In order to search for the core units in a hidden layer, inputs in $D_o^c$ are fed into the model batch by batch. Each batch of inputs tries to find a set of hidden units that preserves the performance of the model w.r.t. the current batch to the best. This set of units consists of those selected by previous batches and also those newly selected ones. To estimate the performance of a group of units on making prediction as class $c$, we keep these units activated and mask all others, and then measure how much performance varies from the old one. We denote the new model with units masked as $F_m(\cdot)$, with the subscript $m$ standing for "masked". The new prediction scores are transformed into a scalar ranging from 0 to 1, denoted as $y_n^b$. The new decision is made by comparing $y_n^b$ with a threshold value 0.5: if $y_n^b \geq 0.5$, the input is given a label of $p_n' = 1$; otherwise $p_n' = 0$. We are interested in comparing label $y_n'$ with the new prediction $p_n'$ (predictions of $F(\cdot)$ and $F_m(\cdot)$). If $y_n' = p_n'$, we say the prediction holds[1] even when we disable certain hidden units, and otherwise the prediction changes. Higher prediction holding rate for class $c$ indicates that this group of units gives better performance.

We model the procedure of searching for the core units in hidden layers as an Indian buffet process (Ghahramani and Griffiths, 2006), and derive the objective function via MAP asymptotic technique (Broderick et al., 2013). Consider the dataset $D_o^c = \{(x_n, y_n')\}_{n=1}^{N'}$, where $y_n'$ is a binary scalar. We use $K$ to denote the number of units to be selected and let $z_{nk} = 1$ if the unit $k$ is selected for making prediction over the datum $x_n$ and 0 otherwise. In the IBP, the dishes taken by customers correspond to units selected for predicting each datum. When predicting the first datum $x_1$, $K_1 \sim Poisson(\gamma)$ units are sampled, where $\gamma > 0$ is a hyper-parameter. Circularly, when predicting the $n$-th datum $x_n$, $\sum_{m=1}^{n-1} K_m$ units are sampled for previous predictions. We use $S_{n-1,k}$ to represent the times for unit $k$ to be selected by the first $n - 1$ times of predictions. Then, the prediction of $x_n$ selects unit $k$ with probability $\frac{S_{n-1,k}}{n}$ and samples $K_n \sim Poisson(\gamma/n)$ new units. To encourage each time of prediction to choose more units that are chosen before, the value of $\gamma$ should not be too large. Totally, for predicting the whole dataset $D_o^c = \{(x_n, y_n')\}_{n=1}^{N'}$, $N'$ inputs sample K units. We form a matrix $Z$ with elements as $z_{nk}$, to indicate the allocation of units chosen by data. The probability of the matrix $Z$ produced by this process is (Ghahramani and Griffiths, 2006)

$$P(Z) = \frac{\gamma^K \exp(-\sum_{n=1}^{N'} \gamma/n)}{\prod_{n=1}^{N'} K_n} \prod_{k=1}^K S_{N',k}^{-1} \binom{N'}{S_{N',k}}^{-1}$$

For each datum $x_n$, we use the function $f_k(x_n)$ to represent the prediction score $y_n^b$ of $x_n$ when only keeping unit $k$ activated. We specify a prior on the function $f_k(x_n)$, $f_k(x_n) \overset{iid}{\sim} N(0, \rho_n^2 I_D)$ for some parameter $\rho_n^2$. Then, the prediction score $y_n^b$ can be expressed as $\sum_{k=1}^K z_{nk} f_k(x_n)$ under a normal distribution with variance $\sigma^2 I_D$ for some parameter $\sigma^2$. In this case, we have the likelihood

---

[1]For the dataset $D_o^c$, we define the prediction holding rate as the number of inputs, whose predictions hold unchanged, divided by the total number of inputs in dataset $D_o^c$

$$P(Y'|Z,F) = \frac{1}{(2\pi\sigma^2)^{N'D/2}} \exp\left\{-\frac{(Y'-\mathbf{tr}(ZF))^\top(Y'-\mathbf{tr}(ZF))}{2\sigma^2}\right\}$$

where $Y'$ is the vector with the $n$-th element as $y'_n$. $F$ is an $\hat{K} \times N'$ matrix with the element $F_{kn}$ as $f_k(x_n)$, where $\hat{K}$ is the number of all units in the hidden layer. We use MAP estimate to determine the number of units $K$, the matrix $Z$ and the matrix $F$ via minimizing $-\log P(Y', Z, F)$. The limit of $\sigma^2$ is set to be 0 asymptotically, so that the prediction score would be centered at a fixed point. We choose some constant $\alpha > 0$ and set $\gamma = \exp(-\alpha/(2\sigma^2))$. Then, with the limit of $\sigma^2 \to 0$, implementing MAP estimate is equivalent to solving the following optimization problem:

$$\underset{K,Z,F}{\operatorname{argmin}}(Y'-\mathbf{tr}(ZF))^\top(Y'-\mathbf{tr}(ZF)) + \alpha \cdot K$$

Here, we can use the vector $Y^b$ to replace $ZF$ in the objective function, whose $n$-th element is the prediction score $y_n^b$. In this case, the objective function can be expressed as Eqn.(1). In practice, due to the limit of the value of $\gamma$, $Z$ is encouraged to be an $N' \times \hat{K}$ matrix with $K$ columns of elements as 1 and others as 0. It means that all the data choose the same units in the hidden layer. Consider the first term in Eqn. (1), it represents the quadratic loss function between prediction scores and binary labels. Thus, the original problem is transformed to a binary optimization problem, in which we select the least number of units that holds the prediction in highest rate.

$$\underset{K,Z}{\operatorname{argmin}} L = (Y'-Y_n^b)^\top(Y'-Y_n^b) + \alpha \cdot K \tag{1}$$

### 3.2 Center-Based Binary Classifier

The output of the classifier $f(\cdot)$ is an $H$-dimensional vector giving prediction scores for each class, which needs to be transformed into a scalar $y_n^b$ for binary optimization. We give a simple binary classifier $f_b(\cdot)$, as Eqn. (2). If $y_n^b \geq 0.5$, the classifier will predict the input vector as class $c$ and $p'_n$ will be assigned 1.

$$y_n^b = 2 \cdot \frac{1}{1+\exp(v_n^b)}$$

$$v_n^b = w_b \cdot \frac{(y_{n,max}^f - r_n)}{\sqrt{\sigma^2(y_n^f)}} \tag{2}$$

In Eqn. (2), $r_n$ and $y_{n,max}^f$ are respectively the $c$-th element and the maximum element of the prediction scores vector $y_n^f$, so $y_{n,max}^f - r_n \geq 0$. As shown in Fig. 1, the $r_n$ term is the prediction score of class $c$, regarded as a center. The distance between maximum element and this center, $(y_{n,max}^f - r_n)$, determines the value of $y_n^b$. In the case that the maximum element of $y_n^f$ is exactly the $c$-th element, the numerator will be zero and $y_n^b$ achieves its maximum value of 1, indicating that the prediction is definitely supposed to be class $c$. On the other hand, if $y_{n,max}^f > r_n$, with a reasonable value of weight $w_b$, $y_n^b$ will be assigned a value smaller than 0.5. The value of weight $w_b$ can be obtained by gradient descent method during binary optimization of unit selection.

### 3.3 Selecting Units in Hidden Layers through Binary Optimization

As shown in Eqn. (1), selecting units in a hidden layer is a typical binary optimization problem, which is difficult to solve because of its NP-hardness. In general, the binary optimization problem can be expressed as Eqn. (3), where $b_n$ is a binary variable and $N^u$ is the number of all binary variables. The minimization objective is a discrete function, so we cannot solve this optimization problem with gradient descent method. Two types of methods are commonly used to solve binary optimization problem, branch and cut method (Mezentsev, 2016; Leontev, 2007; Mezentsev, 2017) and continuous relaxation method (Dai et al., 2016; Giannessi and Tardella, 1998; Lucidi and Rinaldi, 2010). The basic idea of branch and cut method is to build a binary tree and cut off the

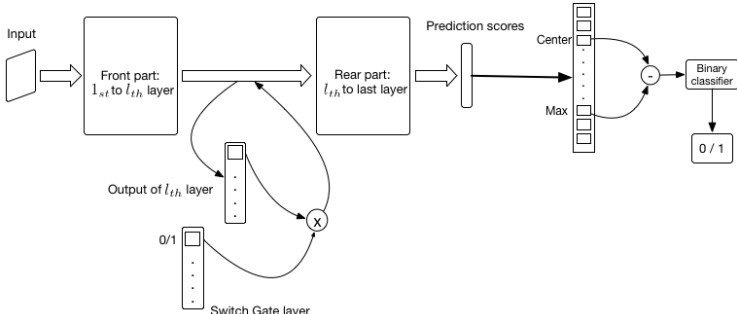

Figure 1: Illustration on how we modify the deep classification model by introducing Switch layers and Binary classifier. A Switch Gate layer is cascaded between $l_{th}$ layer and its next layer. Each unit in the Switch Gate layer works as an ON/OFF gate, by which the output of each feature map in the $l_{th}$ layer is multiplied with 0/1. At the rightmost, the binary classifier transforms multi-dimensional prediction scores into a scalar within range of $[0, 1]$.

enumeration with poor performance. However, in the worst case, this method needs to solve convex sub-problems for $2^n$ time. The other method relaxes the binary constraint to continuous constraint, and adds a penalty term in the objective function to push variables close to zero or one. In the end, results of variables will be rounded to zero or one.

$$\min \quad L(b_1, b_2, \ldots, b_{N^u}), \quad s.t. \quad b_i \in \{0, 1\} \tag{3}$$

To solve the binary optimization problem of selecting units, we cascade a Switch Gate layer between the layer of interest and its next layer. As shown in Fig. 1, every unit in the Switch Gate layer plays the role of an On/Off gate with value of zero or one. The output of each unit in the hidden layer is multiplied with gate value $b_i$. Following the trajectory of continuous relaxation, we transform the binary optimization problem into a new problem, shown in Eqn. (4), where $\phi(b_i) = \sigma(-10 * (b_i - 0.5)^2)$. The function $\phi(b_i)$ is utilized as the penalty of continuous relaxation. The value of $\phi(b_i)$ achieves maximum when $b_i = 0.5$, and falls to zero when $b_i$ gets close to one or zero. With appropriate penalty coefficient $\lambda$, the continuous optimization problem should be equivalent to original optimization problem.

$$\min \quad L(b_1, b_2, \ldots, b_{N^u}) + \sum \lambda \cdot \phi(b_i)$$

$$s.t. \quad b_i \in [0, 1], \quad b_i = \sigma(w \cdot x_g), w > 0 \tag{4}$$

Instead of directly implementing optimization on a gate value $b_i$, we assign a sigmoid function to $b_i$ with input of $x_g$. The value of sigmoid function $b_i$ changes within the range of $(0, 1)$. With augment $w$ positively large enough, the sigmoid function works approximately as a binary step function. A small perturbation of $x_g$ near 0 could push the value from zero to one or inversely. Because the objective function is differentiable with respect to all variables, we can train the Switch Gate layer with gradient descent method. For selecting units in multiple layers simultaneously, each layer of interest could be attached with a Switch Gate layer.

## 4 EXPERIMENT SETUP

The MNIST dataset and the CIFAR10 Dataset are adopted in our experiments, on which two CNN classifiers are well-trained respectively. The classifiers' structures follow those released on Tensor-Flow Official Website[2]. We only connect a Switch Gate layer between the layer of interest and its next layer, and concatenate a binary classifier following the output.

---

[2]The structure of the MNIST classifier is from `https://www.tensorflow.org/get_started/mnist/pros` and the structure of the CIFAR10 classifier is from `https://www.tensorflow.org/tutorials/deep_cnn`

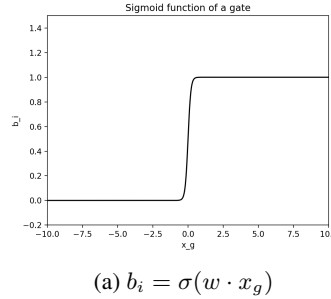
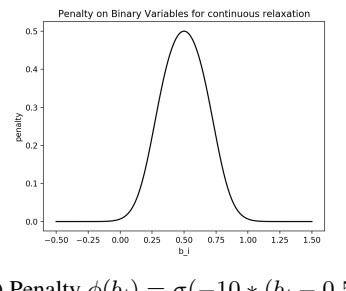

(a) $b_i = \sigma(w \cdot x_g)$  (b) Penalty $\phi(b_i) = \sigma(-10 * (b_i - 0.5)^2)$

Figure 2: Value curves of the function of a switch gate in the switch Gate layer (a) and the penalty function for continuous relaxation (b).

**Structures:** As shown in Fig. 3 (a), the classifier on MNIST dataset consists of six hidden layers: two Conv layers (conv1, conv2), both followed by a max-pooling layer (max1, max2), and two fully-connected layers (fc1, fc2). The conv1 layer has 32 feature maps[3] and the conv2 layer has 64 feature maps. The output of the last fully-connected layer is a 10-dimensional vector with prediction scores for each class. This network is trained by minimizing softmax cross entropy between prediction scores and labels of input images. During training, the drop rate is set to be 0.5. Finally, this network achieves 99.2% test accuracy.

The structure of CIFAR10 classifier is shown in Fig. 3 (b), composed of two convolutional layers (conv1, conv2) followed by three fully-connected layers (fc1 ,fc2, softmax-linear). These two convolutional layers both have 64 feature maps. The output of each convolutional layer, before being passed to the next layer with weights, is processed by a max-pooling layer and a layer for local response normalization. The output of the last fully-connected layer is a 10-dimensional vector giving prediction scores. The training images are cropped to $24 \times 24$ pixels, randomly for training and centrally for evaluation. After training, the test accuracy achieves 86.5%.

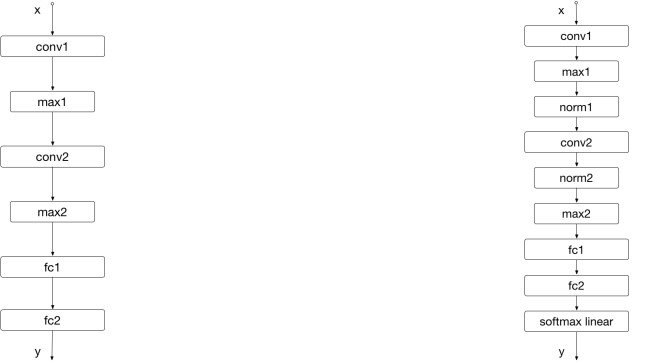

(a) Structure of a MNIST classifier  (b) Structure of a CIFAR10 classifier

Figure 3: Structures of two classifiers used for experiments. (a) Structure of the MNIST classifier, which consists of 2 convolutional layers, each followed by 1 max-pooling layer, and 2 fully-connected layers. (b) Structure of the CIFAR10 classifier, which contains 2 convolutional layers, each followed with 1 max-pooling layer and 1 local response normalization layer, and 3 fully-connected layers.

**Layers of interest:** For the MNIST classifier, we investigate the two max-pooling layers and the input layer. The max-pooling layers down-sample the output of previous convolutional layers unit-wise. Selecting units in a max-pooling layer requires a less number of parameters than the previous convolutional layer. When selecting units in the max1 layer, a Switch Gate layer with 32 units is connected between layer max1 and layer conv2. A binary classifier is attached to the output score

---

[3]In convolutional layers, a unit refers to a feature map

vector to transform a 10-dimensional vector into a binary scalar, indicating whether the prediction is the class of interest. We train the Switch Gate layer by solving the problem in Eqn. (4). At the end of optimization, we obtain a Switch Gate layer with all unit values around zero or one. Following a similar procedure, a Switch layer with 64 units is connected between layer max2 and fc1 to select units in the max2 layer. In addition to layer-wise experiments, we also attempt to select units in max1 layer and max2 layer simultaneously. We connect a Switch Gate layer following the two layers respectively. In the input layer, with our method we can obtain a highlight map that shows which pixels are most responsible for predictions of a particular class. For the CIFAR10 classifier, we select units in two max-pooling layers. Both two max-pooling layers consist of 64 feature maps. Switch Gate layers will be connected between each of two max-pooling layers and its next layer.

## 5 RESULT ANALYSIS

### 5.1 RESULTS ON MNIST

Firstly, we attach a Switch Gate layer following the max1 layer in the MNIST classifier. We implement binary optimization for all 10 classes. By choosing appropriate coefficients in the loss function, we keep the prediction holding rate nearly 99.5% and obtain the least number of units that keep most information for predicting. The binary optimization is implemented three times for every class. For each time, the dataset $D_o^c$ is randomly permuted. As shown in Table. 1, the number of units obtained varies within a small range ($\leq 2$) for three times of optimization. The small variance of units number might be due to two aspects: a little change in prediction holding rate and the information redundancy of a hidden layer. Though we use the same coefficients in the loss function for each time of optimization, the prediction holding rate keeps within a small range around 99.5%. A small increase in prediction holding rate may require keeping one or two more units. On the other hand, to evince the redundancy in the max1 layer, we record the positions of units obtained for five times of optimization for some classes, as shown in Fig. 4. We find that despite a similar number of units obtained, the positions of units are not fixed. It can be inferred that the number of units obtained indicates the amount of information, and in order for high prediction holding rate, the model just needs to retain the enough amount of information. The number of units required for a particular amount of information could vary within a small range. We also implement binary opti-

(a) Results of Layer max1

| CoI | Optimization | Holding Rate(%) | Num of Units |
|---|---|---|---|
| 0 | 1 | 99.7 | 11 |
| | 2 | 99.5 | 11 |
| | 3 | 99.7 | 12 |
| 1 | 1 | 99.5 | 11 |
| | 2 | 99.7 | 13 |
| | 3 | 99.6 | 13 |
| 2 | 1 | 99.7 | 17 |
| | 2 | 99.6 | 15 |
| | 3 | 99.3 | 15 |
| 3 | 1 | 99.4 | 12 |
| | 2 | 99.7 | 12 |
| | 3 | 99.4 | 14 |
| 4 | 1 | 99.6 | 13 |
| | 2 | 99.2 | 14 |
| | 3 | 99.7 | 14 |
| 5 | 1 | 99.6 | 14 |
| | 2 | 99.6 | 14 |
| | 3 | 99.4 | 14 |
| 6 | 1 | 99.3 | 9 |
| | 2 | 99.4 | 9 |
| | 3 | 99.4 | 10 |
| 7 | 1 | 99.8 | 15 |
| | 2 | 99.6 | 13 |
| | 3 | 99.6 | 14 |
| 8 | 1 | 99.5 | 16 |
| | 2 | 99.6 | 15 |
| | 3 | 99.7 | 16 |
| 9 | 1 | 99.6 | 15 |
| | 2 | 99.4 | 14 |
| | 3 | 99.3 | 16 |

(b) Results of Layer max2

| CoI | Optimization | Holding Rate(%) | Num of Units |
|---|---|---|---|
| 0 | 1 | 99.6 | 12 |
| | 2 | 99.7 | 12 |
| | 3 | 99.6 | 13 |
| 1 | 1 | 99.5 | 12 |
| | 2 | 99.6 | 11 |
| | 3 | 99.5 | 12 |
| 2 | 1 | 99.5 | 12 |
| | 2 | 99.4 | 10 |
| | 3 | 99.4 | 12 |
| 3 | 1 | 99.2 | 15 |
| | 2 | 99.5 | 16 |
| | 3 | 99.4 | 16 |
| 4 | 1 | 99.6 | 17 |
| | 2 | 99.5 | 17 |
| | 3 | 99.5 | 15 |
| 5 | 1 | 99.5 | 15 |
| | 2 | 99.5 | 15 |
| | 3 | 99.4 | 14 |
| 6 | 1 | 99.5 | 12 |
| | 2 | 99.6 | 13 |
| | 3 | 99.5 | 12 |
| 7 | 1 | 99.7 | 14 |
| | 2 | 99.7 | 13 |
| | 3 | 99.5 | 12 |
| 8 | 1 | 99.4 | 16 |
| | 2 | 99.2 | 17 |
| | 3 | 99.4 | 18 |
| 9 | 1 | 99.3 | 13 |
| | 2 | 99.5 | 15 |
| | 3 | 99.4 | 15 |

Table 1: Results of binary optimization in max1 layer and max2 layer of MNIST classifier. For each CoI (class of interest), optimization of unit selection was implemented three times. The holding rates and number of units are recorded in tables above.

mization on units in the max2 layer and get similar observations. However, we compare the number of units in max2 with that in max1 for the same class, and find that the max1 layer retains nearly 30% to 50% units while the max2 layer only 17% to 30%. One of the most important functions of a convolutional filter is to capture useful features for classification. Units in higher hidden layers capture more complex features and information. It does make sense that the max2 layer only needs 17% to 30% to hold correct predictions. Furthermore, we evaluate the effectiveness of our method on selecting units in a series of layers at the same time. We attach Switch Gate layers following the max1 layer and the max2 layer. With binary optimization, units in two layers are selected with prediction holding rate above 99.2%. As shown in Table. 2, however, the numbers of units in two layers are not consistent. Because of the redundancy in hidden layers, there are several choices of unit selection to keep prediction holding rate nearly 99.2%.

| CoI | Optimization | Holding Rate(%) | Num of Units: max1 | Num of Units: max2 |
|---|---|---|---|---|
| 0 | 1 | 99.4 | 14 | 14 |
| | 2 | 99.2 | 11 | 13 |
| | 3 | 99.2 | 14 | 12 |
| 1 | 1 | 99.3 | 13 | 16 |
| | 2 | 99.4 | 13 | 12 |
| | 3 | 99.4 | 14 | 12 |
| 8 | 1 | 99.2 | 21 | 17 |
| | 2 | 99.4 | 25 | 24 |
| | 3 | 99.3 | 21 | 20 |

Table 2: Results of selecting units in max1 and max2 layers simultaneously. For the well-trained MNSIT classifier, we selected units in the max1 layer and max2 layer at the same time. We list results of experiments for three classes (0, 1, and 8) here.

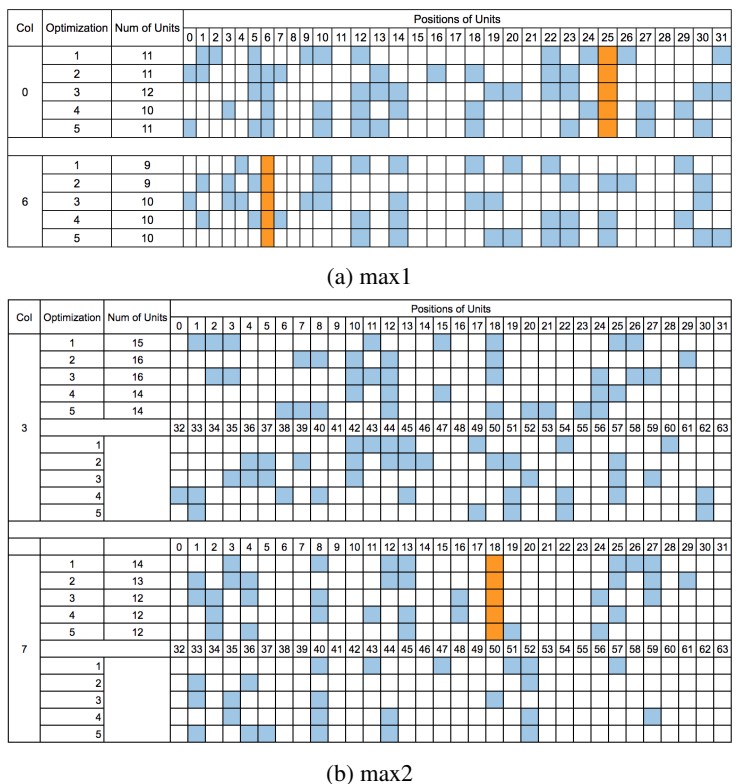

(a) max1

(b) max2

Figure 4: Positions of units selected in max1 and max2 layers of MNIST classifier. For each time of optimization, the positions of units obtained are marked with light blue color. The position with orange color represents appearing for all five times. The layer max1 has 32 feature maps; results of layer max1 are listed in Figure (a). The layer max2 has 64 feature maps; the results are listed in Figure (b).

In order to demonstrate the reasonability of results, after each time of optimization, we sort units obtained in the hidden layer according to their corresponding controlling gate values. Then, we mask units one by one, from those with small gate values to those with large gate values, and calculate corresponding holding rates. Fig. 6a shows that the prediction holding rate drops fast with the decrease of the number of units, which indicates that the number of units obtained is the least in order for keeping the holding rate nearly 99.5%. Besides, we randomly select the same number of units from the layer of interest, masking all other units, and make prediction for the optimization dataset. We iterate this procedure for ten times and get holding rates in the range of 47% to 98% that are smaller than the holding rate obtained via binary optimization.

Applying the proposed method on the input layer, we can get the informative features in the pixel space for each class. We only need to modify the optimization dataset for each class, removing those inputs that are not predicted by the original classifier as this class. In other words, we only use "positive" data to formulate this binary optimization problem and exert an implicit bias for the optimization. We visualize informative pixels by plotting a figure with values of controlling gates in the Switch Gate layer, as shown in Fig. 5, where points highlighted in the figures contribute most to the prediction of the corresponding class. The informative features we find are compatible for a whole class of images.

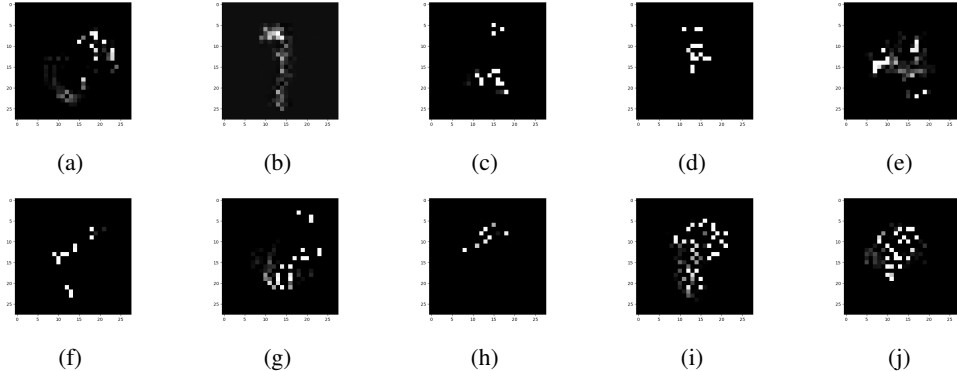

(a)  (b)  (c)  (d)  (e)

(f)  (g)  (h)  (i)  (j)

Figure 5: Visualization of informative features in pixel-space for the MNIST classifier. Informative features for class 0 to 4 (from left to right) are listed in the first row, and those for class 5 to 9 (from left to right) are listed in the second row. The highlight points in figures are those most contributed to the prediction of corresponding class.

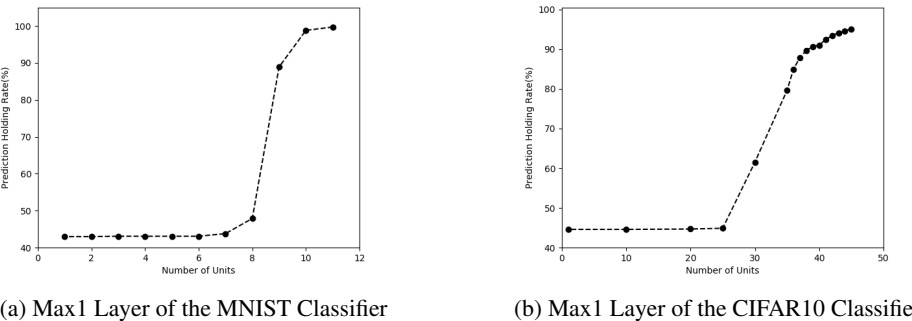

(a) Max1 Layer of the MNIST Classifier  (b) Max1 Layer of the CIFAR10 Classifier

Figure 6: The prediction holding rate vs number of units. (a): After obtaining the number and positions of units for class 0 in max1 layer in the MNSIT classifier, we modify the original model with all other units in max1 layer masked and get a model with prediction holding rate near 99.5%. We decrease the number of units kept and calculate corresponding holding rates. (b): We select the units in max1 layer for class airplane in the CIFAR10 classifier and do the same procedure as before.

(a) Results of Layer max1

| CoI | Optimization | Holding Rate(%) | Num of Units |
|---|---|---|---|
| 0 | 1 | 95.2 | 47 |
|  | 2 | 95.0 | 45 |
|  | 3 | 95.1 | 45 |
| 1 | 1 | 96.9 | 42 |
|  | 2 | 97.8 | 42 |
|  | 3 | 97.4 | 41 |
| 2 | 1 | 95.4 | 52 |
|  | 2 | 96.2 | 54 |
|  | 3 | 95.6 | 53 |
| 3 | 1 | 94.0 | 50 |
|  | 2 | 93.0 | 48 |
|  | 3 | 95.5 | 51 |
| 4 | 1 | 96.5 | 48 |
|  | 2 | 95.8 | 50 |
|  | 3 | 96.4 | 50 |
| 5 | 1 | 94.6 | 46 |
|  | 2 | 94.2 | 46 |
|  | 3 | 95.1 | 47 |
| 6 | 1 | 96.3 | 44 |
|  | 2 | 95.8 | 43 |
|  | 3 | 95.8 | 43 |
| 7 | 1 | 95.3 | 43 |
|  | 2 | 94.7 | 44 |
|  | 3 | 95.5 | 42 |
| 8 | 1 | 97.0 | 41 |
|  | 2 | 96.5 | 44 |
|  | 3 | 96.7 | 42 |
| 9 | 1 | 96.8 | 44 |
|  | 2 | 96.2 | 42 |
|  | 3 | 97.0 | 44 |

(b) Results of Layer max2

| CoI | Optimization | Holding Rate(%) | Num of Units |
|---|---|---|---|
| 0 | 1 | 97.3 | 46 |
|  | 2 | 97.0 | 43 |
|  | 3 | 97.5 | 44 |
| 1 | 1 | 97.2 | 30 |
|  | 2 | 97.6 | 31 |
|  | 3 | 97.9 | 32 |
| 2 | 1 | 96.6 | 44 |
|  | 2 | 97.5 | 47 |
|  | 3 | 97.0 | 45 |
| 3 | 1 | 96.3 | 51 |
|  | 2 | 94.8 | 49 |
|  | 3 | 96.5 | 50 |
| 4 | 1 | 96.6 | 40 |
|  | 2 | 96.6 | 39 |
|  | 3 | 96.8 | 39 |
| 5 | 1 | 97.0 | 46 |
|  | 2 | 96.7 | 46 |
|  | 3 | 96.6 | 46 |
| 6 | 1 | 97.2 | 41 |
|  | 2 | 97.2 | 38 |
|  | 3 | 97.3 | 38 |
| 7 | 1 | 97.5 | 41 |
|  | 2 | 96.6 | 40 |
|  | 3 | 96.7 | 40 |
| 8 | 1 | 97.7 | 35 |
|  | 2 | 97.0 | 34 |
|  | 3 | 97.1 | 34 |
| 9 | 1 | 97.3 | 35 |
|  | 2 | 97.1 | 34 |
|  | 3 | 96.3 | 34 |

Table 3: Results of binary optimization in max1 layer and max2 layer of CIFAR10 classifier. For each CoI (class of interest), optimization of unit selection was implemented three times. The holding rates and number of units are recorded in tables above

| CoI | Optimization | Holding Rate(%) | Num of Units: max1 | Num of Units: max2 |
|---|---|---|---|---|
| airplane | 1 | 93.4 | 42 | 47 |
|  | 2 | 92.7 | 41 | 44 |
|  | 3 | 94.1 | 47 | 43 |
| automobile | 1 | 96.5 | 46 | 39 |
|  | 2 | 96.0 | 44 | 39 |
|  | 3 | 95.3 | 41 | 36 |
| bird | 1 | 93.0 | 47 | 46 |
|  | 2 | 93.2 | 52 | 48 |
|  | 3 | 93.5 | 48 | 51 |

Table 4: Results of selecting units in max1 and max2 layers simultaneously. In the CIFAR10 classifier, we selected units in the max1 layer and max2 layer at the same time. We list results of experiments for three classes (airplane, automobile and bird) here.

## 5.2 RESULTS ON CIFAR10

The second series of experiments are conducted on CIFAR10 dataset, which contains tens of thousands of RGB images with ten classes, such as airplane, dog, ship, etc. In Table 3, we use the same number to represent each class as the original labels do. Binary optimization is implemented in the first maxpooling layer (max1) and the second one (max2). Layer max1 and layer max2 both have 64 feature maps. We choose coefficients in the loss function to keep the prediction holding rate nearly 95%. If we put too much weight on holding rate, almost all units should be retained. Unlike results of the MNIST classifier, the numbers of units kept in max1 layer and max2 layer are similar. For some class, the number of units obtained in the max2 layer is a little smaller than that in the max1 layer, but still over 30. We can explain this difference from the perspective of network capacity $vs$ dataset complexity. Although the CIFAR10 classifier is deeper than the MNIST classifier and has more units in hidden layers, it only achieves 86.5% test accuracy, less than 99.2% of the MNIST classifier, due to larger complexity of RGB images in CIFAR10 data set. Units in the second layer could not extract structured and complete features. Thus, the second layer has to retain enough units and feed enough information to the following network for correct prediction. We implement five times of optimization and compare the positions of units obtained. The number of units for five times of optimization varies within a small range. However, similar with the observations on the MNIST classifier, the positions of units are not fixed. We also perform experiments on selecting units in max1 and max2 layer at the same time, listing results in Table 4, and obtain

a group of forward channels for each class with prediction holding rate above 93%. It can be seen that our method can effectively find the core hidden units in the CIFAR10 classifier, but due to the information redundancy in hidden layers, positions of units obtained are not fixed.

## 6    CONCLUSION

In this work we proposed a method to investigate the effects of units in hidden layers on prediction making through finding the core units that make most contributions to the prediction. Our proposed deep model interpreter enables a better understanding of DNN models, which is beneficial to their related research and application. Experiments on MNIST and CIFAR10 datasets demonstrated its effectiveness. Furthermore, this method can be used for informative pixels visualization via applying binary optimization on the input layer.

In future work, we may follow this work and explore in two aspects. First, since the number of units obtained via binary optimization varies within a small range, but the positions of units are not fixed, we want to know what common characteristics these groups of units share. Second, since the proposed method returns a model with the trivial units masked, which holds predictions for a particular class in a high rate, then we want to see whether its usage can be extended to shrinking model size or accelerating computation.

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
