# OpenReview forum: "Interpreting Deep Classification Models With Bayesian Inference"
_ICLR.cc/2018/Conference — Reject_

### Official Review · AnonReviewer1 · 2017-11-28
**The paper proposes to identify the core units of a deep neural network in a one-vs-remaining manner. It is unclear what is the use of identifying these core units.**

**Rating:** 3
**Confidence:** 3

**Review:**

The paper intends to interpret a well-trained multi-class classification deep neural network by discovering the core units of one or multiple hidden layers for prediction making. However, these discovered core units are specific to a particular class, which are retained to maintain the deep neural network’s ability to separate that particular class from the other ones. Thus, these non-core units for a particular class could be core units for separating another class from the remaining ones. Consequently, the aggregation of all class-specific core units could include all hidden units of a layer. Therefore, it is hard for me to understand what’s the motivation to identify the core units in a one-vs-remaining manner. At this moment, these identified class-specific core units are useful for neither reducing the size of the network, nor accelerating computation.

---

### Official Review · AnonReviewer2 · 2017-11-28
**Hard to follow some parts; more analysis needed.**

**Rating:** 5
**Confidence:** 3

**Review:**

Pros
- The paper proposes a novel formulation of the problem of finding hidden units
  that are crucial in making a neural network come up with a certain output.
- The method seems to be work well in terms of isolating a few hidden units that
  need to be kept while preserving classification accuracy.

Cons
- Sections 3.1 and 3.2 are hard to understand. There seem to be inconsistencies
  in the notation. For example,
(1) It would help to clarify whether y^b_n is the prediction score or its
transformation into [0, 1]. The usage is inconsistent.
(2) It is not clear how "y^b_n can be expressed as \sum_{k=1}^K z_{nk}f_k(x_n)"
in general. This is only true for the penultimate layer, and when y^b_n denotes
the input to the output non-linearity. However, this analysis seems to be
applied for any hidden layer and y^b_n is the output of the non-linearity unit
("The new prediction scores are transformed into a scalar ranging from 0 to 1,
denoted as y^b_n.")
(3) Section 3.1 denotes the DNN classifier as F(.), but section 3.2 denotes the
same classifier as f(.).
(4) Why is r_n called the "center" ? I could not understand in what sense is
this the center, and of what ? It seems that the max value has been subtracted
from all the logits into a softmax (which is a fairly standard operation).

- The analysis seems to be about finding neurons that contribute evidence for
  a particular class. This does not address the issue of understanding why the
network makes a certain prediction for a particular input. Therefore this
approach will be of limited use.

- The paper should include more analysis of how this method helps interpret the
  actions of the neural net, once the core units have been identified.
Currently, the focus seems to be on demonstrating that the classifier
performance is maintained as a significant fraction of hidden units are masked.
However, there is not enough analysis on showing whether and how the identified
hidden units help "interpret" the model.

Quality
The idea explored in the paper is interesting and the experiments are described
in enough detail. However, the writing still needs to be polished.

Clarity
The problem formulation and objective function (Section 3.1) was hard to follow.

Originality
This approach to finding important hidden units is novel.

Significance
The paper addresses an important problem of trying to have more interpretable
neural networks. However, it only identifies hidden units that are important for
a class, not what are important for any particular input.  Moreover, the main
thesis of the paper is to describe a method that helps interpret neural network
classifiers. However, the experiments only focus on identifying important hidden
units and fall short of actually providing an interpretation using these hidden
units.

---

### Official Review · AnonReviewer3 · 2017-12-04

**Rating:** 3
**Confidence:** 4

**Review:**

The paper develops a technique to understand what nodes in a neural network are important
for prediction. The approach they develop consists of using an Indian Buffet Process
to model a binary activation matrix with number of rows equal to the number of examples.
The binary variables are estimated by taking a relaxed version of the
asymptotic MAP objective for this problem. One question from the use of the
Indian Buffet Process: how do the asymptotics of the feature allocation determine
the number of hidden units selected?

Overall, the results didn't warrant the complexity of the method. The results are neat, but
I couldn't tell why this approach was better than others.

Lastly, can you intuitively explain the additivity assumption in the distribution for p(y')

---

### Decision · Program_Chairs · 2018-01-29
**ICLR 2018 Conference Acceptance Decision**

**Decision:**

Reject

**Comment:**

The paper proposes a new method for interpreting the hidden units of neural networks by employing an Indian Buffet Process. The reviewers felt that the approach was interesting, but at times hard to follow and more analysis was needed. In particular, it was difficult to glean any advantage of this method over others. The authors did not provide a response to the reviews.